# Federated Linear Bandits
# with Finite Adversarial Actions

**Li Fan**
University of Virginia
lf2by@virginia.edu

**Ruida Zhou**
Texas A&M University
ruida@tamu.edu

**Chao Tian**
Texas A&M University
chao.tian@tamu.edu

**Cong Shen**
University of Virginia
cong@virginia.edu

## Abstract

We study a federated linear bandits model, where $M$ clients communicate with a central server to solve a linear contextual bandits problem with finite adversarial action sets that may be different across clients. To address the unique challenges of *adversarial finite* action sets, we propose the `FedSupLinUCB` algorithm, which extends the principles of SupLinUCB and OFUL algorithms in linear contextual bandits. We prove that `FedSupLinUCB` achieves a total regret of $\tilde{O}(\sqrt{dT})$, where $T$ is the total number of arm pulls from all clients, and $d$ is the ambient dimension of the linear model. This matches the minimax lower bound and thus is order-optimal (up to polylog terms). We study both asynchronous and synchronous cases and show that the communication cost can be controlled as $O(dM^2 \log(d) \log(T))$ and $O(\sqrt{d^3 M^3} \log(d))$, respectively. The `FedSupLinUCB` design is further extended to two scenarios: (1) variance-adaptive, where a total regret of $\tilde{O}(\sqrt{d \sum_{t=1}^{T} \sigma_t^2})$ can be achieved with $\sigma_t^2$ being the noise variance of round $t$; and (2) adversarial corruption, where a total regret of $\tilde{O}(\sqrt{dT} + dC_p)$ can be achieved with $C_p$ being the total corruption budget. Experiment results corroborate the theoretical analysis and demonstrate the effectiveness of `FedSupLinUCB` on both synthetic and real-world datasets.

## 1  Introduction

In the canonical formulation of contextual bandits, a single player would repeatedly make arm-pulling decisions based on contextual information with the goal of maximizing the long-term reward. With the emerging *federated learning* paradigm (McMahan et al., 2017) where multiple clients and a server jointly learn a global model with each client locally updating the model with its own data and server only aggregating the local models periodically, researchers have started exploring contextual bandits algorithms in such federated learning setting (Dubey and Pentland, 2020; Huang et al., 2021; Li and Wang, 2022a,b). This federated contextual bandits framework broadens the applicability of contextual bandits to practical scenarios such as recommender systems, clinical trials, and cognitive radio. In these applications, although the goal is still to maximize the cumulative reward for the overall system, decision-making and observations are naturally distributed at the participating clients.

Several intrinsic challenges arise with the federated contextual bandit formulation. One important issue is that besides regret, we should also take into account the communication cost, which is usually the system bottleneck. To reduce the communication cost while maintaining the same regret guarantee, the clients should transmit the necessary information to the server only when the local

37th Conference on Neural Information Processing Systems (NeurIPS 2023).

information has accumulated to the extent that it would affect the decision-making. Compared with the centralized contextual bandits, which have a linearly growing communication cost, algorithms for federated contextual bandits attempt to achieve a comparable regret with sub-linear communication cost.

Second, most existing studies on federated contextual bandits focus on the synchronous communication scenario (Huang et al., 2021; Li and Wang, 2022b), in which all participating clients first upload local information and then download updated global information from the server in each communication round. This stringent communication requirement is often not met in practice. A recent work of Li and Wang (2022a) studies the asynchronous federated linear bandit problem. However, communications for different clients are not independent in their approach because the upload from one client may trigger the server to perform downloads for all clients. To address this issue, He et al. (2022a) proposes FedLinUCB, which enables independent synchronizations between clients and the server.

Third, the majority of prior studies on federated linear bandits focused on the infinite-arm setting (Li and Wang, 2022b,a; He et al., 2022a) (see Section 2 for a detailed literature review). From a methodology point of view, these papers largely build on the OFUL principle (Abbasi-Yadkori et al., 2011). One notable exception is Huang et al. (2021), which studies synchronous communication with fixed contexts and proposes the Fed-PE algorithm based on the phased elimination G-optimal design (Lattimore and Szepesvári, 2020). To the best of our knowledge, no prior result exists for federated linear bandits with finite arms and time-evolving adversarial contexts, which is the focus of our work.

Table 1: Comparison of this paper with related works

| System | Action | Algorithm | Regret | Communication |
|---|---|---|---|---|
| Single-player | infinite arm | OFUL (Abbasi-Yadkori et al., 2011) | $d\sqrt{T}\log T$ | N/A |
| Single-player | finite fixed arm | PE + G-optimal (Lattimore and Szepesvári, 2020) | $O(\sqrt{dT}\log T)$ | N/A |
| Single-player | finite adversarial arm | SupLinUCB (Chu et al., 2011) | $O(\sqrt{dT}\log^3 T)$ | N/A |
| Federated (Async) | infinite arm | FedLinUCB (He et al., 2022a) | $O(d\sqrt{T}\log T)$ | $O(dM^2\log T)$ |
| Federated (Async) | infinite arm | Async-LinUCB (Li and Wang, 2022a) | $O(d\sqrt{T}\log T)$ | $O(dM^2\log T)$ |
| Federated (Sync) | infinite arm | DisLinUCB (Wang et al., 2019) | $O(d\sqrt{T}\log^2 T)$ | $O(dM^{3/2})$ |
| Federated (Sync) | finite fixed arm | Fed-PE (Huang et al., 2021) | $O(\sqrt{dT}\log T)$ | $O(d^2MK\log T)$ |
| Federated (Async) | finite adversarial arm | FedSupLinUCB (This work) | $O(\sqrt{dT}\log^3 T)$ | $O(dM^2\log d\log T)$ |
| Federated (Sync) | finite adversarial arm | FedSupLinUCB (This work) | $O(\sqrt{dT}\log^3 T)$ | $O(d^{3/2}M^{3/2}\log(d))$ |

$d$: the dimension of the unknown parameter, $M$: the number of clients, $K$: the number of finite actions, $T$: the total arm pulls from all clients.

**Main contributions.** Our main contributions are summarized as follows.

- We develop a general federated bandits framework, termed `FedSupLinUCB`, for solving the problem of federated linear contextual bandits with finite adversarial actions. `FedSupLinUCB` extends SupLinUCB (Chu et al., 2011; Ruan et al., 2021) and OFUL (Abbasi-Yadkori et al., 2011), two important principles in (single-player, centralized) linear bandits, to the federated bandits setting with a carefully designed layered successive screening.

- We instantiate `FedSupLinUCB` with both asynchronous and synchronous client activities. For the former setting, we propose Async-`FedSupLinUCB` where communication is triggered only when the cumulative local information impacts the exploration uncertainty to a certain extent. We prove that Async-`FedSupLinUCB` achieves $\tilde{O}(\sqrt{dT})$ regret with $O(dM^2\log d\log T)$ communication cost, which not only reduces the regret by $\sqrt{d}$ compared with previous results on asynchronous federated linear bandits with infinite arms, but also matches the minimax lower bound up to polylog terms, indicating that Async-`FedSupLinUCB` achieves order-optimal regret.

- For synchronous communications, we propose Sync-`FedSupLinUCB`, which has a refined communication design where only certain layers are communicated, as opposed to the complete information. Sync-`FedSupLinUCB` achieves order-optimal regret of $\tilde{O}(\sqrt{dT})$ with horizon-independent communication cost $O(\sqrt{d^3M^3}\log d)$. Compared with the best previous result (Huang et al., 2021) which achieves the same order-optimal regret but only for *fixed* actions, we show that it is the *finite* actions that fundamentally determines the regret behavior in the federated linear bandits setting.

- We further develop two extensions of `FedSupLinUCB`: (1) Variance-adaptive `FedSupLinUCB`, for which a total regret of $\tilde{O}(\sqrt{d\sum_{t=1}^{T}\sigma_t^2})$ is achieved, where $\sigma_t^2$ is the noise variance at round $t$. (2)

Adversarial corruption `FedSupLinUCB`, for which a total regret of $\tilde{O}(\sqrt{dT} + dC_p)$ is achieved, where $C_p$ is the total corruption budget.

## 2 Related Works

The linear bandit model, as a generalization of finite armed bandits with linear contextual information, has been extensively studied. The setting of infinite arm sets solved by LinUCB was analyzed in (Dani et al., 2008; Abbasi-Yadkori et al., 2011), which achieves regret $\tilde{O}(d\sqrt{T})$ with appropriate confidence width (Abbasi-Yadkori et al., 2011) and matches the lower bound (Dani et al., 2008) up to logarithmic factors. In contrast, algorithms like SupLinRel (Auer, 2002) and SupLinUCB (Chu et al., 2011) achieve $\tilde{O}(\sqrt{dT})$ in the setting of finite time-varying adversarial arm sets under $K \ll 2^d$, with a lower bound $\Omega(\sqrt{dT})$ (Chu et al., 2011). The SupLinUCB algorithm was later optimized and matches the lower bound up to iterated logarithmic factors in Li et al. (2019). As a special case of the finite arm setting, if the arm set is time-invariant, an elimination-based algorithm (Lattimore and Szepesvári, 2020) via G-optimal design can be applied to achieve similar optimal performance.

The federated linear bandits problems were studied under the settings of infinite arm set (Dubey and Pentland, 2020; Li et al., 2020; Li and Wang, 2022a) and time-invariant finite arm set (Huang et al., 2021), while the time-varying finite arm set setting has not been well explored. A finite time-varying arm set has many meaningful practical applications such as recommendation system (Li et al., 2010; Chu et al., 2011), and the distributed (federated) nature of the applications naturally falls in the federated linear bandits problem with finite time-varying arms. The paper fills this gap by generalizing the SupLinUCB algorithm to the federated setting.

We study both the asynchronous setting (Li and Wang, 2022a) (He et al., 2022a), where clients are active on their own and full participation is not required, and the synchronous setting (Shi et al., 2021; Dubey and Pentland, 2020), where all the clients make decisions at each round and the communication round requires all the clients to upload new information to the server and download the updated information. We design algorithms so as to reduce the communication cost while maintaining optimal regret. Technically, the communication cost is associated with the algorithmic adaptivity, since less adaptivity requires fewer updates and thus fewer communication rounds. The algorithmic adaptivity of linear bandits algorithms was studied in the single-player setting (Han et al., 2020) (Ruan et al., 2021). It was also considered in the federated setting (Wang et al., 2019; Huang et al., 2021; Salgia and Zhao, 2023).

## 3 System Model and Preliminaries

### 3.1 Problem Formulation

We consider a federated linear contextual bandits model with $K$ finite but possibly time-varying arms. The model consists of $M$ clients and one server in a star-shaped communication network. Clients jointly solve a linear bandit problem by collecting local information and communicating with the central server through the star-shaped network in a federated manner, with no direct communications among clients. The only function of the server is to aggregate received client information and to send back updated information to clients. It cannot directly play the bandits game.

Specifically, some clients $I_t \subseteq [M]$ are active at round $t$. Client $i \in I_t$ receives $K$ arms (actions to take) associated with contexts $\{x_{t,a}^i\}_{a \in [K]} \subset \mathbb{R}^d$ with $\|x_{t,a}^i\|_2 \leq 1$. Here we adopt the oblivious adversarial setting, where all contexts are chosen beforehand, and not dependent on previous game observation. Client $i$ then pulls an arm $a_t^i \in [K]$ based on the information collected locally as well as previously communicated from the server. A reward $r_{t,a_t^i}^i = \theta^\top x_{t,a_t^i}^i + \epsilon_t$ is revealed privately to client $i$, where $\theta \in \mathbb{R}^d$ is an unknown weight vector with $\|\theta\|_2 \leq 1$ and $\epsilon_t$ is an independent 1-sub-Gaussian noise. At the end of round $t$, depending on the communication protocol, client $i$ may exchange the collected local information with the server so that it can update the global information.

We aim to design algorithms to guide the clients' decision-making and overall communication behaviors. We analyze two patterns of client activity. 1) **Synchronous**: all $M$ clients are active at each round. 2) **Asynchronous**: one client is active at each round. For the latter case, we further assume that client activity is independent of data and history. Denote by $T_i$ the number of times client

$i$ is active. In the former case, $T_i = T_j, \forall i, j \in [M]$, while in the latter case, $T_i$ may be different among clients. We define $T = \sum_{i=1}^{M} T_i$ as the total number of arm pulls from all clients.

The performance is measured under two metrics – *total regret* and *communication cost*, which concern the decision-making effectiveness and the communication efficiency respectively. Denote by $P_T^i = \{t \in [T] \mid i \in I_t\}$ the set of time indices at which client $i$ is active, with $|P_T^i| = T_i$. The total regret is defined as

$$R_T = \sum_{i=1}^{M} R_T^i = \sum_{i=1}^{M} \mathbb{E} \left[ \sum_{t \in P_T^i} r_{t,a_t^{i,*}}^i - r_{t,a_t^i}^i \right], \tag{1}$$

where $a_t^{i,*} = \arg\max_{a \in [K]} \theta^\top x_{t,a}^i$. We define the communication cost as the total number of communication rounds between clients and the server.

### 3.2 Preliminaries

**Information encoding.** In the linear bandits setting (federated or not), the information a client acquires is usually encoded by the gram matrix and the action-reward vector. Specifically, when the client has observed $n$ action-reward pairs $\{(x_t, r_t)\}_{t=1}^n$, the information is encoded by matrix $A_n = \sum_{t=1}^n x_t x_t^\top$ and vector $b_n = \sum_{t=1}^n r_t x_t$. Denote by $\text{Encoder}(\cdot)$ this encoding function, i.e., $A_n, b_n \leftarrow \text{Encoder}(\{x_t, r_t\}_{t=1}^n)$.

**Communication criterion.** Communication in our proposed framework is data-dependent, in the same spirit as the "doubling trick" introduced in Abbasi-Yadkori et al. (2011) to reduce the computation complexity in single-player linear bandits. The key idea is that communication is triggered only when the cumulative local information, represented by the determinant of the gram matrix $A_n$, affects the exploration uncertainty to a great extent and hence the client needs to communicate with the server. Detailed communication protocols will be presented in each algorithm design.

**Synchronization procedure.** Denote by $\text{Sync}()$ a routine that $n$ clients (client 1, ..., client $n$) first communicate their local gram matrices and action-reward vectors to the server, and the server then aggregates the matrices (and vectors) into one gram matrix (and action-reward vector) and transmits them back to the $n$ clients. Specifically, each client $i$ holds newly observed local information $(\Delta A^i, \Delta b^i)$, which is the difference between the client's current information $(A^i, b^i)$ and the information after the last synchronization. In other words, $(\Delta A^i, \Delta b^i)$ is the information that has not been communicated to the server. The server, after receiving the local information $\{(\Delta A^i, \Delta b^i)\}_{i=1}^n$, updates the server-side information $(A^{ser}, b^{ser})$ by $A^{ser} \leftarrow A^{ser} + \sum_{i=1}^n \Delta A^i, b^{ser} \leftarrow b^{ser} + \sum_{i=1}^n \Delta b^i$ and sends them back to each of the $n$ clients. Each client $i$ will then update the local information by $A^i \leftarrow A^{ser}, b^i \leftarrow b^{ser}$. The procedure is formally presented in Algorithm 1.

---

**Algorithm 1** $\text{Sync}(s, \text{server}, \text{client } 1, \dots \text{client } n)$

---
1: **for** $i = 1, 2, \dots, n$ **do**                    ▷ Client-side local information upload
2:     Client $i$ sends the local new layer $s$ information $(\Delta A_s^i, \Delta b_s^i)$ to the server
3: **end for**
4: Update server's layer $s$ information:      ▷ Server-side information aggregation and distribution

$$A_s^{ser} \leftarrow A_s^{ser} + \sum_{i=1}^n \Delta A_s^i, \quad b_s^{ser} \leftarrow b_s^{ser} + \sum_{i=1}^n \Delta b_s^i$$

5: Send server information $A_s^{ser}, b_s^{ser}$ back to all clients
6: **for** $i = 1, 2, \dots, n$ **do**
7:     $A_s^i \leftarrow A_s^{ser}, b_s^i \leftarrow b_s^{ser}, \Delta A_s^i \leftarrow 0, \Delta b_s^i \leftarrow 0$      ▷ Client $i$ updates the local information
8: **end for**

---

## 4 The `FedSupLinUCB` Framework

In this section, we present a general framework of federated bandits for linear bandits with finite oblivious adversarial actions. Two instances (asynchronous and synchronous) of this general framework will be discussed in subsequent sections.

**Building block: SupLinUCB.** As the name suggests, the proposed `FedSupLinUCB` framework is built upon the principle of SupLinUCB (Chu et al., 2011; Ruan et al., 2021). The information $(A, b)$ is useful in the sense that the reward corresponding to an action $x$ can be estimated within confidence interval $x^\top \hat{\theta} \pm \alpha \|x\|_{A^{-1}}$, where $\hat{\theta} = A^{-1}b$. It is shown in Abbasi-Yadkori et al. (2011) that in linear bandits (even with an infinite number of actions) with $\alpha = \tilde{O}(\sqrt{d})$, the true reward is within the confidence interval with high probability. Moreover, if the rewards in the action-reward vector $b$ are mutually independent, $\alpha$ can be reduced to $O(1)$. The former choice of $\alpha$ naturally guarantees $\tilde{O}(d\sqrt{T})$ regret. However, to achieve regret $\tilde{O}(\sqrt{dT})$, it is critical to keep $\alpha = O(1)$. This is fulfilled by the SupLinUCB algorithm (Chu et al., 2011) and then recently improved by Ruan et al. (2021). The key intuition is to successively refine an action set that contains the optimal action, where the estimation precision of sets is geometrically strengthened. Specifically, the algorithm maintains $(S + 1)$ layers of information pairs $\{(A_s, b_s)\}_{s=0}^{S}$, and the rewards in the action-reward vectors are mutually independent, except for layer 0. The confidence radius for each layer $s$ is $w_s = 2^{-s}d^{1.5}/\sqrt{T}$.

---

**Algorithm 2** S-LUCB

---

1: **Initialization**: $S = \lceil \log d \rceil, \overline{w}_0 = d^{1.5}/\sqrt{T}, \overline{w}_s \leftarrow 2^{-s}\overline{w}_0, \forall s \in [1 : S]$.
2: $\alpha_0 = 1 + \sqrt{d \ln(2M^2 T/\delta)}, \alpha_s \leftarrow 1 + \sqrt{2\ln(2KMT \ln d/\delta)}, \forall s \in [1 : S]$
3: **Input**: Client $i$ (with local information $A^i, b^i, \Delta A^i, \Delta b^i$), contexts set $\{x_{t,1}^i, \dots, x_{t,K}^i\}$
4: $A_{t,s}^i \leftarrow A_s^i + \Delta A_s^i, b_{t,s}^i \leftarrow b_s^i + \Delta b_s^i$ or $A_{t,s}^i \leftarrow A_s^i, b_{t,s}^i \leftarrow b_s^i$ for `lazy update`
5: $\hat{\theta}_s \leftarrow (A_{t,s}^i)^{-1}b_{t,s}^i, \hat{r}_{t,s,a}^i = \hat{\theta}_s^\top x_{t,a}^i, w_{t,s,a}^i \leftarrow \alpha_s \|x_{t,a}^i\|_{(A_{t,s}^i)^{-1}}, \forall s \in [0 : S], \forall a \in [K]$
6: $s \leftarrow 0; \mathcal{A}_0 \leftarrow \{a \in [K] \mid \hat{r}_{t,0,a}^i + w_{t,0,a}^i \geq \max_{a \in [K]}(\hat{r}_{t,0,a}^i - w_{t,0,a}^i)\}$     ▷ Initial screening
7: **repeat**     ▷ Layered successive screening
8:     **if** $s = S$ **then**
9:         Choose action $a_t^i$ arbitrarily from $\mathcal{A}_S$
10:     **else if** $w_{t,s,a}^i \leq \overline{w}_s$ for all $a \in \mathcal{A}_s$ **then**
11:         $\mathcal{A}_{s+1} \leftarrow \{a \in \mathcal{A}_s \mid \hat{r}_{t,s,a}^i \geq \max_{a' \in \mathcal{A}_s}(\hat{r}_{t,s,a'}^i) - 2\overline{w}_s\}; s \leftarrow s + 1$
12:     **else**
13:         $a_t^i \leftarrow \arg\max_{\{a \in \mathcal{A}_s, w_{t,s,a}^i > \overline{w}_s\}} w_{t,s,a}^i$
14:     **end if**
15: **until** action $a_t^i$ is found
16: Take action $a_t^i$ and and receive reward $r_{t,a_t^i}^i$
17: $\Delta A_s^i \leftarrow \Delta A_s^i + x_{t,a_t^i}^i x_{t,a_t^i}^{i\top}, \Delta b_s^i \leftarrow \Delta b_s^i + r_{t,a_t^i}^i x_{t,a_t^i}^i$     ▷ Update local information
18: **Return** layer index $s$

---

**FedSupLinUCB.** S-LUCB, presented in Algorithm 2, combines the principles of SupLinUCB and OFUL (Abbasi-Yadkori et al., 2011) and is the core subroutine for `FedSupLinUCB`. We maintain $S = \lceil \log d \rceil$ information layers, and the estimation accuracy starts from $d^{1.5}/\sqrt{T}$ of layer 0 and halves as the layer index increases. Finally, it takes $\Theta(\log d)$ layers to reach the sufficient accuracy of $\sqrt{d/T}$ and achieves the minimax-optimal regret.

When client $i$ is active, the input parameters $(A^i, b^i)$ contain information received from the server at the last communication round, and $(\Delta A^i, \Delta b^i)$ is the new local information collected between two consecutive communication rounds. $\{x_{t,1}^i, \dots, x_{t,K}^i\}$ is the set of contexts observed in this round. Client $i$ can estimate the unknown parameter $\theta$ either with all available information or just making a lazy update. This choice depends on the communication protocol and will be elaborated later. During the decision-making process, client $i$ first makes arm elimination at layer 0 to help bootstrap the accuracy parameters. Then, it goes into the layered successive screening in the same manner as the SupLinUCB algorithm, where we sequentially eliminate suboptimal arms depending on their empirical means and confidence widths. After taking action $a_t^i$ and receiving the corresponding reward $r_{t,a_t^i}^i$, client $i$ updates its local information set $(\Delta A_s^i, \Delta b_s^i)$ by aggregating the context into layer $s$ in which we take the action, before returning layer $s$.

# 5 Asynchronous `FedSupLinUCB`

In the asynchronous setting, only one client is active in each round. Note that global synchronization and coordination are not required, and all inactive clients are idle.

## 5.1 Algorithm

We first initialize the information for all clients and the server (gram matrix and action-reward vector) in each layer $s \in [0:S]$. We assume only one client $i_t$ is active at round $t$. It is without loss of generality since if multiple clients are active, we can queue them up and activate them in turn. More discussion of this equivalence can be found in He et al. (2022a); Li and Wang (2022a). The active client chooses the action, receives a reward, updates local information matrices of layer $s$ with a lazy update according to S-LUCB, and decides whether communication with the server is needed by the criterion in Line 7 of Algorithm 3. If communication is triggered, we synchronize client $i_t$ with the server by Algorithm 1.

---

**Algorithm 3** Async-FedSupLinUCB

1: **Initialization**: $T$, $C$, $S = \lceil \log d \rceil$
2: $\{A_s^{ser} \leftarrow I_d, b_s^{ser} \leftarrow 0 \mid s \in [0:S]\}$             ▷ Server initialization
3: $\{A_s^i \leftarrow I_d, \Delta A_s^i, b_s^i, \Delta b_s^i \leftarrow 0 \mid s \in [0:S], i \in [M]\}$     ▷ Clients initialization
4: **for** $t = 1, 2, \cdots, T$ **do**
5:      Client $i_t = i$ is active, and observes $K$ contexts $\{x_{t,1}^i, x_{t,2}^i, \cdots, x_{t,K}^i\}$
6:      $s \leftarrow$ S-LUCB $\left(\text{client } i, \{x_{t,1}^i, x_{t,2}^i, \cdots, x_{t,K}^i\}\right)$ with lazy update
7:      **if** $\frac{\det\left(A_s^i + \Delta A_s^i\right)}{\det(A_s^i)} > (1 + C)$ **then**
8:          Sync($s$, server, clients $i$) for each $s \in [0:S]$
9:      **end if**
10: **end for**

---

## 5.2 Performance Analysis

**Theorem 5.1.** *For any $0 < \delta < 1$, if we run Algorithm 3 with $C = 1/M^2$, then with probability at least $1 - \delta$, the regret of the algorithm is bounded as $R_T \leq \tilde{O}\left(\sqrt{d \sum_{i=1}^M T_i}\right) = \tilde{O}\left(\sqrt{dT}\right)$. Moreover, the corresponding communication cost is bounded by $O(dM^2 \log d \log T)$.*

*Remark 1.* The minimax lower bound of the expected regret for linear contextual bandits with $K$ adversarial actions is $\Omega(\sqrt{dT})$, given in Chu et al. (2011). Theorem 5.1 indicates that Async-FedSupLinUCB achieves order-optimal regret (up to polylog term) with $O(dM^2 \log d \log T)$ communication cost. To the best of our knowledge, this is the first algorithm that achieves the (near) optimal regret in federated linear bandits with finite adversarial actions.

*Remark 2.* Without any communication, each client would execute SupLinUCB (Chu et al., 2011) for $T_i$ rounds locally, and each client can achieve regret of order $\tilde{O}(\sqrt{dT_i})$. Therefore, the total regret of $M$ clients is upper bound by $R_T \leq \sum_{i=1}^M \sqrt{dT_i} \, \text{polylog}(T) \leq \sqrt{dM \sum_{i=1}^M T_i} \, \text{polylog}(T)$, where the last inequality becomes equality when $T_i = T_j, \forall i, j \in [M]$. Compared with conducting $M$ independent SupLinUCB algorithms locally, Async-FedSupLinUCB yields an average *per-client gain of* $1/\sqrt{M}$, demonstrating that communications in the federated system can speed up local linear bandits decision-making at clients.

*Remark 3.* Most previous federated linear bandits consider the infinite action setting, based on the LinUCB principle (Abbasi-Yadkori et al., 2011). Async-FedSupLinUCB considers a finite adversarial action setting and has a $\sqrt{d}$ reduction on the regret bound. Fed-PE proposed in Huang et al. (2021) also considers the finite action setting. However, their action sets are fixed. We generalize their formulation and take into account a more challenging scenario, where the finite action set can be chosen adversarially. The regret order is the same as Fed-PE (ignoring the ploylog term), indicating

that it is the *finite* actions as opposed to *fixed* actions that fundamentally leads to the $\sqrt{d}$ regret improvement in the federated linear bandits setting.

**Communication cost analysis of** `FedSupLinUCB`. We sketch the proof for the communication cost bound in Theorem 5.1 in the following, while deferring the detailed proofs for the regret and the communication cost to Appendix C.

We first study the communication cost triggered by some layer $s$. Denote by $A_{t,s}^{ser}$ the gram matrix in the server aggregated by the gram matrices uploaded by all clients up to round $t$. Define $T_{n,s} = \min\{t \in [T]| \det(A_{t,s}^{ser}) \geq 2^n\}$, for each $n \geq 0$. We then divide rounds into epochs $\{T_{n,s}, T_{n,s} + 1, \cdots, \min(T_{n+1,s} - 1, T)\}$ for each $n \geq 0$. The number of communications triggered by layer $s$ within any epoch can be upper bounded by $2(M + 1/C)$ (see Lemma C.1), and the number of non-empty epochs is at most $d\log(1 + T/d)$ by Lemma A.1. Since there are $S = \lceil \log d \rceil$ layers and synchronization among all layers is performed once communication is triggered by any layer (Line 8 in Algorithm 3), the total communication cost is thus upper-bounded by $O(d(M + 1/C) \log d \log T)$. Plugging $C = 1/M^2$ proves the result.

We note that although choosing a larger $C$ would trigger fewer communications, the final choice of $C = 1/M^2$ takes into consideration both the regret and the communication cost, i.e., to achieve a small communication cost while maintaining an order-optimal regret.

# 6   Synchronous `FedSupLinUCB`

In the synchronous setting, all clients are active and make decisions at each round. Though it can be viewed as a special case of the asynchronous scenario (clients are active and pulling arms in a round-robin manner), the information update is broadcast to all clients. In other words, the key difference from the asynchronous scenario besides that all clients are active at each round is that when a client meets the communication criterion, *all* clients will upload local information to the server and download the updated matrices. This leads to a higher communication cost per communication round, but in this synchronous scenario, knowing all clients are participating allows the communicated information to be well utilized by other clients. This is in sharp contrast to the asynchronous setting, where if many other clients are active in the current round, uploading local information to the clients seems unworthy. To mitigate the total communication cost, we use a more refined communication criterion to enable time-independent communication cost.

## 6.1   The Algorithm

The Sync-FedSupLinUCB algorithm allows each client to make decisions by the `S-LUCB` subroutine. Note that the decision-making is based on all available local information instead of the lazy update in the Async-FedSupLinUCB algorithm. The communication criterion involves the count of rounds since the last communication, which forces the communication to prevent the local data from being obsolete. Some layers may trigger the communication criterion either because the local client has gathered enough new data or due to having no communication with the server for too long. We categorize these layers in the CommLayers and synchronize all the clients with the server.

## 6.2   Performance Analysis

**Theorem 6.1.** *For any $0 < \delta < 1$, if we run Algorithm 4 with $D = \frac{T_c \log T_c}{d^2 M}$, with probability at least $1 - \delta$, the regret of the algorithm is bounded as $R_T \leq \tilde{O}(\sqrt{dMT_c})$ where $T_c$ is the total per-client arm pulls. Moreover, the corresponding communication cost is bounded by $O(\sqrt{d^3 M^3} \log d)$.*

*Remark 4.* Theorem 6.1 demonstrates Sync-FedSupLinUCB also achieves the minimax regret lower bound while the communication cost is independent of $T_c$. It is particularly beneficial for large $T_c$. Especially, the number of total rounds in the synchronous scenario is $T = MT_c$, while in the asynchronous setting, we have $T = \sum_{i=1}^{M} T_i$ rounds.

**Communication cost analysis of** `Sync-FedSupLinUCB`. We sketch the proof for the communication cost bound in Theorem 6.1 below, while deferring the detailed proofs for the regret and the communication cost to Appendix D.

---

**Algorithm 4** Sync-FedSupLinUCB

---

1: **Initialization**: $T_c, D, S = \lceil \log d \rceil, t_{last}^s \leftarrow 0, \forall s \in [0:S]$, CommLayers $\leftarrow \emptyset$.
2: $\{A_s^{ser} \leftarrow I_d, b_s^{ser} \leftarrow 0 \mid s \in [0:S]\}$             $\triangleright$ Server initialization
3: $\{A_s^i \leftarrow I_d, \Delta \tilde{A}_s^i, b_s^i, \Delta b_s^i \leftarrow 0 \mid s \in [0:S], i \in [M]\}$      $\triangleright$ Clients initialization
4: **for** $t = 1, 2, \cdots, T_c$ **do**
5:      **for** $i = 1, 2, \cdots, M$ **do**
6:          Client $i_t = i$ is active, and observes $K$ contexts $\{x_{t,1}^i, x_{t,2}^i, \cdots, x_{t,K}^i\}$
7:          $s \leftarrow$ S-LUCB $\left(\text{client } i, \{x_{t,1}^i, x_{t,2}^i, \cdots, x_{t,K}^i\}\right)$
8:          **if** $(t - t_{last}^s) \log \frac{\det\left(A_s^i + \Delta A_s^i\right)}{\det(A_s^i)} > D$ **then**
9:             Add $s$ to CommLayers
10:          **end if**
11:      **end for**
12: **end for**
13: **for** $s \in$ CommLayers **do**
14:      Sync($s$, server, clients $[M]$); $t_{last}^s \leftarrow t$, CommLayers $\leftarrow \emptyset$
15: **end for**

---

We call the chunk of consecutive rounds without communicating information in layer $s$ (except the last round) an *epoch*. Information in layer $s$ is collected locally by each client and synchronized at the end of the epoch, following which the next epoch starts. Denoted by $A_{p,s}^{all}$ the synchronized gram matrix at the end of the $p$-th epoch. For any value $\beta > 0$, there are at most $\lceil \frac{T_c}{\beta} \rceil$ epochs that contain more than $\beta$ rounds by pigeonhole principle. If the $p$-th epoch contains less than $\beta$ rounds, then $\log\left(\frac{\det\left(A_{p,s}^{all}\right)}{\det\left(A_{p-1,s}^{all}\right)}\right) > \frac{D}{\beta}$ based on the communication criterion and the fact that $\sum_{p=1}^P \log \frac{\det\left(A_{p,s}^{all}\right)}{\det\left(A_{p-1,s}^{all}\right)} \le R_s = O(d \log(T_c))$ (see Equation (6)). The number of epochs containing rounds fewer than $\beta$ is at most $O(\lceil \frac{R_s}{D/\beta} \rceil)$. Noting that $D = \frac{T_c \log(T_c)}{d^2 M}$, the total number of epochs for layer $s$ is at most $\lceil \frac{T_c}{\beta} \rceil + \lceil \frac{R_s \beta}{D} \rceil = O(\sqrt{\frac{T_c R_s}{D}}) = O(\sqrt{d^3 M})$ by taking $\beta = \sqrt{\frac{D T_c}{R_s}}$. The total communication cost is thus upper bounded by $O(SM\sqrt{d^3 M}) = O(\log(d)\sqrt{d^3 M^3})$.

# 7 Extensions of FedSupLinUCB

In this section, we extend the FedSupLinUCB algorithm to address two distinct settings in federated systems: scenarios characterized by heterogeneous variances, and those affected by adversarial corruptions.

## 7.1 Federated Heteroscedastic Linear Bandits

We have so far focused on the federated linear bandits with 1-sub-Gaussian reward noises. In this section, we adapt Async-FedSupLinUCB to the case where the reward noises have *heterogeneous* variances, which extends the *heteroscedastic linear bandits* as studied in Zhou et al. (2021); Zhou and Gu (2022) to the asynchronous federated setting, where one client is active at a time. Specifically, the reward noises $\{\epsilon_t\}_{t \in [T]}$ are independent with $|\epsilon_t| \le R, \mathbb{E}[\epsilon_t] = 0$ and $\mathbb{E}[\epsilon_t^2] \le \sigma_t^2$, where $\sigma_t$ is known to the active client.

We propose a variance-adaptive Asyc-FedSupLinUCB and analyze its regret and the communication cost in the theorem below, with the algorithm and the proof details in Appendix E due to space constraint. The regret is significantly less than that of the Async-FedSupLinUCB when the variances $\{\sigma_t^2\}$ are small.

**Theorem 7.1.** *For any $0 < \delta < 1$, if we run the variance-adaptive Async-FedSupLinUCB algorithm in Appendix E with $C = 1/M^2$, with probability at least $1 - \delta$, the regret is bounded as $R_T \le \tilde{O}(\sqrt{d \sum_{t=1}^T \sigma_t^2})$, and the communication cost is bounded by $O(dM^2 \log^2 T)$.*

## 7.2 Federated Linear Bandits with Corruption

We further explore asynchronous federated linear bandits with adversarial corruptions, where an adversary inserts a corruption $c_t$ to the reward $r_t$ of the active client at round $t$. The total corruption is bounded by $\sum_{t=1}^{T} |c_t| \leq C_p$. We incorporate the idea of linear bandits with adversarial corruption studied in He et al. (2022b) to the proposed `FedSupLinUCB` framework and propose the Robust Async-`FedSupLinUCB` algorithm, with details in Appendix F. Robust Async-`FedSupLinUCB` can achieve the optimal minimax regret (matching the lower bound in He et al. (2022b)) while incurring a low communication cost.

**Theorem 7.2.** *For any $0 < \delta < 1$, if we run the Robust Async-`FedSupLinUCB` algorithm in Appendix F with $C = 1/M^2$, with probability at least $1 - \delta$, the regret is bounded as $R_T \leq \tilde{O}(\sqrt{dT} + dC_p)$, and the communication cost is bounded by $O(dM^2 \log d \log T)$.*

## 8 Experiments

We have experimentally evaluated `FedSupLinUCB` in the asynchronous and synchronous settings on both synthetic and real-world datasets. We report the results in this section.

### 8.1 Experiment Results Using Synthetic Dataset

We simulate the federated linear bandits environment specified in Section 3. With $T = 40000$, $M = 20$, $d = 25$, $\mathcal{A} = 20$, contexts are uniformly randomly sampled from an $l_2$ unit sphere, and reward $r_{t,a} = \theta^\top x_{t,a} + \epsilon_t$, where $\epsilon_t$ is Gaussian distributed with zero mean and variance $\sigma = 0.01$. It should be noted that while $M$ clients participate in each round in the synchronous scenario, only one client is active in the asynchronous case. In the plots, the $x$-axis coordinate denotes the number of arm pulls, which flattens the actions in the synchronous setting.

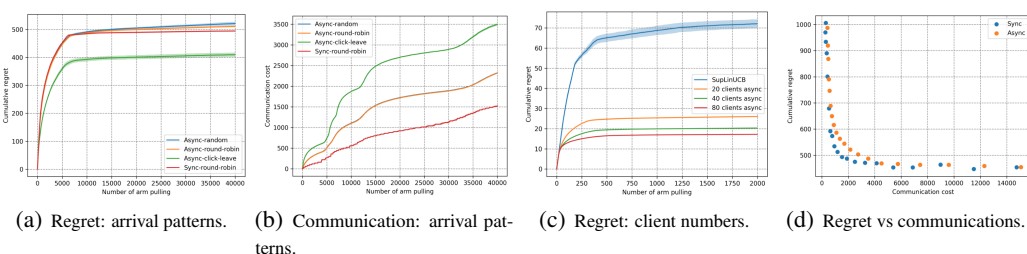

(a) Regret: arrival patterns.    (b) Communication: arrival patterns.    (c) Regret: client numbers.    (d) Regret vs communications.

Figure 1: Experimental results with the synthetic dataset.

**Arrival pattern.** We first investigate the impact of different arrival patterns (the sequence of activating clients): (1) **Random**, which randomly allocates $T/M$ arm pulls in $[T]$ for each client. (2) **Round-robin**, i.e. $[1, 2, 3, \cdots, M, 1, 2, 3, \cdots M, \cdots]$. (3) **Click-leave**, i.e. $[1, 1, \cdots, 2, 2, \cdots, \cdots, M, M, \cdots]$. The regret and the communication cost of these three arrival patterns in the synthetic experiment are reported in Figure 1(a) and Figure 1(b), respectively. We note that although the upper bound analysis in our proof is for the worst-case instance, the numerical results suggest that different arrival patterns result in diverse regret performances. Round-robin and random patterns are more challenging since both local bandit learning and each client's policy updates happen relatively slowly. The click-leave pattern, which is the closest to the centralized setting, achieves the best regret. In addition, compared with Async-`FedSupLinUCB`, Sync-`FedSupLinUCB` achieves better cumulative regrets with a higher communication cost.

**Amount of clients.** The per-client cumulative regret as a function of $T_c = T/M$ with different amounts of clients is plotted in Figure 1(c). In comparison to the baseline SupLinUCB, `FedSupLinUCB` algorithms achieve better regret via communication between clients and the server. We can see from the experiment that `FedSupLinUCB` significantly reduces the per-client regret compared with SupLinUCB, and achieves a better regret as $M$ increases in both asynchronous and synchronous settings.

**Trade-off between regrets and communications.** We evaluate the tradeoff between communication and regret by running `FedSupLinUCB` with different communication threshold values $C$ and $D$ in asynchronous and synchronous settings respectively. The results are reported in Figure 1(d), where each scattered dot represents the communication cost and the cumulative regret that `FedSupLinUCB` has achieved with a given threshold value at round $T = 40000$. We see a clear tradeoff between the regret and the communication. More importantly, Sync-`FedSupLinUCB` achieves a better tradeoff than Async-`FedSupLinUCB`.

## 8.2 Experiment Results Using Real-world Dataset

We further investigate how efficiently the federated linear bandits algorithm performs in a more realistic and difficult environment. We have carried out experiments utilizing the real-world recommendation dataset MovieLens 20M (Harper and Konstan, 2015). Following the steps in Li and Wang (2022b), we first filter the data by maintaining users with above 2500 movie ratings and treating rating points greater than 3 as positive, ending up with $N = 37$ users and 121934 total movie rating interactions. Then, we follow the process described in Cesa-Bianchi et al. (2013) to generate the contexts set, using the TF-IDF feature $d = 25$ and the arm set $K = 20$. We plot the per-client normalized rewards of the FedSupLinUCB algorithm with different client numbers $M$ in synchronous and asynchronous cases respectively. Note that the per-client cumulative rewards here are normalized by a random strategy. From Figure 2(a) and Figure 2(b), we can see that in both synchronous and asynchronous experiments, `FedSupLinUCB` has better rewards than SupLinUCB, and the advantage becomes more significant as the number of users increases.

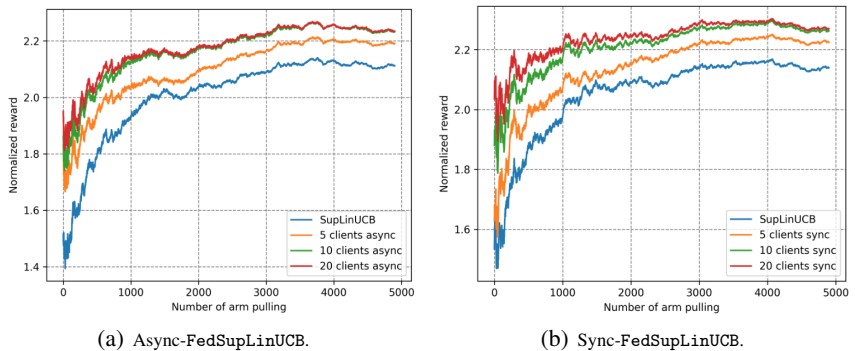

(a) `Async-FedSupLinUCB`.    (b) `Sync-FedSupLinUCB`.

Figure 2: Experimental results with the real-world MovieLens-20M dataset.

## 9 Conclusion

We studied federated linear bandits with finite adversarial actions, a model that has not been investigated before. We proposed `FedSupLinUCB` that extends the SupLinUCB and OFUL principles to the federated setting in both asynchronous and synchronous scenarios, and analyzed their regret and communication cost, respectively. The theoretical results proved that `FedSupLinUCB` is capable of approaching the minimal regret lower bound (up to polylog terms) while only incurring sublinear communication costs, suggesting that it is the *finite* actions that fundamentally determines the regret behavior in the federated linear bandits setting. Furthermore, we examined the extensions of the algorithm design to the variance-adaptive and adversarial corruption scenarios.

## Acknowledgments and Disclosure of Funding

The work of LF and CS was supported in part by the U.S. National Science Foundation (NSF) under grants 2143559, 2029978, and 2132700.

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
