# OpenReview forum: "Federated Linear Bandits with Finite Adversarial Actions"
_NeurIPS.cc/2023/Conference — NeurIPS 2023 poster_

### Official Review · Reviewer_64z5 · 2023-07-05

**Soundness:** 3 good
**Presentation:** 3 good
**Contribution:** 2 fair
**Rating:** 6
**Confidence:** 3

**Summary:**

This paper studies the linear contextual bandits problem with federated learning of $M$ clients communicating with a central sever. In particular, the paper assumes adversarial finite action, and considers two cases of the communication: asynchronous and synchronous.

Following the idea of OFUL, this work extends the previous SupLinUCB in linear contextual bandits to deal with the federated learning scenario, and proposes the algoirthm FedSupLinUCB which achieves $\widetilde{O}(\sqrt{dT})$ where $T$ is the total number of pulls (the summation of cumulative pulls over $M$ clients together), and $d$ is the dimension. More importantly, this result is attained with limited communication cost $\mathcal{O}(\sqrt{d^3M^3}\log(d))$ for synchronous case, and $\mathcal{O}(dM^2\log(d)\log(T))$ for asynchronous case.

In addition, the FedSupLinUCB further extends to the variance-adaptive and adversarial corruption scenarios.

**Strengths:**

The paper is clearly-written and well-organized. The notations are well-defined and the meanings are explained clearly prior to their usages.

1. Combining online learning and federated learning is an interesting direction. The proposed algorithm attains the optimal regret bound of such problem up to some $\log T$ factors, and further ensures limited communication cost, which could be dominating in the distributed systems. Among these results, the most interesting one is that for the synchronous case, which is independent with the horizon $T$.

2. The instance-dependent regret bounds for the variance-adaptive and adversarial corruption scenarios are also very interesting. The worst case regret bound is not meaningful in most real world applications.


**Weaknesses:**

This paper does not have any specific weakness.

**Questions:**

1. Any lower bound of the communication cost in the synchronous case or the asynchronous case?
2. Any idea to improve the regret bounds to the optimal regret bound $\mathcal{O}(\sqrt{dT\log T})$, which can be obtained by Fed-PE?



**Limitations:**

This work is pure theoretical, and does not have any negative societal impact.

---

> ### Author Rebuttal · Authors · 2023-08-08
>
> We thank the reviewer for the clear summary and for finding our paper interesting. Please see our response below with respect to your specific comments.
>
> **Q1**: "Any lower bound of the communication cost in the synchronous case or the asynchronous case?"
>
> **Response**: There are some most recent works, which appeared after the submission of our manuscript, focusing on analyzing the communication cost complexity in federated bandit problems. E.g., in [R2], the communication cost is measured in bits in the federated linear bandits setting with a simple unit ball action.
>
> **Q2**: "Any idea to improve the regret bounds to the optimal regret bound $O(\sqrt{dT\log(T)})$, which can be obtained by Fed-PE?"
>
> **Response**: It is important to note that our study differs from Fed-PE, which primarily focuses on finite and fixed context sets. In contrast, we address a finite and time-evolving context setting, presenting new challenges. As such, the G-optimal design method utilized in Fed-PE is not applicable to our context. To the best of our knowledge, our regret is order-optimal in this scenario.
>
> [R2] Salgia, Sudeep, and Qing Zhao. "Distributed linear bandits under communication constraints." International Conference on Machine Learning. PMLR, 2023.

---

> > ### Comment · Reviewer_xbXb · 2023-08-16
> >
> > Thank you for your answer,
> >
> > Regarding Q1: it would be useful to add this concurrent results to the final version of this paper.
> >
> > Regarding Q2: thank you for the clarification. Could you elaborate on what makes you think that your regret is order optimal and how to derive that lower bound?

---

> > > ### Author Response · Authors · 2023-08-16
> > >
> > > Thank you for your message. We appreciate the reviewer's suggestions and will certainly incorporate the new communication results into the final version.
> > >
> > > In single-client linear bandits with finite adversarial actions, it has been shown that an agent incurs regret at least $\Omega(\sqrt{dT})$ [R3], where $d$ is the dimension of the unknown vector and $T$ is the total number of rounds. In the federated linear bandits setting of interests, there are $M$ clients and each runs with $T$ rounds. When communicating arbitrarily, it can be viewed as a single agent that runs a total of $MT$ rounds, which incurs at least $\Omega(\sqrt{dMT})$ regrets, which gives a lower bound for the federated linear bandits with finite actions. The proposed algorithms achieve, $\tilde{O}(\sqrt{d MT})$ by omitting logarithmic factors, and thus are order optimal.
> > >
> > > Note that the proposed algorithms achieve order optimality while maintaining small communication costs, which is the essence of federated learning. A more comprehensive analysis of order optimality and the benefit of the federated linear bandit framework can be found in Remark 2, which follows Theorem 5.1 in our paper.
> > >
> > > [R3] Chu, Wei, et al. "Contextual bandits with linear payoff functions." Proceedings of the Fourteenth International Conference on Artificial Intelligence and Statistics. JMLR Workshop and Conference Proceedings, 2011.

---

> > > > ### Comment · Reviewer_64z5 · 2023-08-17
> > > > **Response to author's rebuttal**
> > > >
> > > > Thank the authors for their response.  The paper could be improved by including these concurrent results. I tend to keep my score.

---

### Official Review · Reviewer_EJUn · 2023-07-06

**Soundness:** 3 good
**Presentation:** 3 good
**Contribution:** 3 good
**Rating:** 5
**Confidence:** 4

**Summary:**

This paper studies a federated linear bandits model, where M clients communicate with a central server to solve a linear contextual bandits problem with finite adversarial action sets and proposes the FedSupLinUCB algorithm, which extends the SupLinUCB and OFUL principles in linear contextual bandits. Both asynchronous and synchronous cases are considered. Experiment results corroborate the theoretical analysis and demonstrate the effectiveness of FedSupLinUCB on both synthetic and real-world datasets.



**Strengths:**

The method description, theoretical derivation and complexity analysis of the article are very detailed and well organized.




**Weaknesses:**

See Questions part.


**Questions:**

1. Do the adversarial corruption actios only exist in the setting of asynchronous case?
2. Are there differences in intensity for adversarial actions? If so, do different intensities have different effects on the experimental scene?
3. Is Robust Async-FedSupLinUCB a general robust method or can only be used against a specific adversarial setting？

---

> ### Author Rebuttal · Authors · 2023-08-08
>
> We thank the reviewer for the interesting questions regarding the proposed FedSupLinUCB algorithm.
>
> **Q1**: "Do the adversarial corruption actions only exist in the setting of the asynchronous case?"
>
> **Response**: The asynchronous case contains the synchronous case as a special case. We thus designed the robust async-FedSupLinUCB algorithm that handles the adversarial corrupted actions in this general asynchronous case, which thus can be directly applied to the synchronous case. Note that in the study without corruption, we treat the synchronous setting separately and designed a more communication-efficient scheme without time horizon dependence.
>
> **Q2**: "Are there differences in intensity for adversarial actions? If so, do different intensities have different effects on the experimental scene?"
>
> **Response**: We measure the total intensity of the adversarial actions by $C_p$ as shown in Lines 281-283, and its impact on the regret is represented by an additional additive term of $d C_p$ in Theorem 8.1. Please correct us if your remark is misunderstood by us. We have made a general assumption on the context set, considering it to be both finite and adversarial. This assumption enhances the practicality of our algorithm, making it applicable to a wide range of real-world scenarios.
>
> **Q3**: "Is Robust Async-FedSupLinUCB a general robust method or can only be used against a specific adversarial setting?"
>
> **Response**: Robust Async-FedSupLinUCB is a general robust method that is capable of handling any additive corruption on the reward function, as long as the total corruption budget is constrained.

---

> > ### Comment · Reviewer_EJUn · 2023-08-13
> > **Response to author's rebuttal**
> >
> > Thanks to the authors' response which has addressed my minor concerns, and I will keep my preliminary rating

---

### Official Review · Reviewer_xbXb · 2023-07-06

**Soundness:** 3 good
**Presentation:** 3 good
**Contribution:** 3 good
**Rating:** 6
**Confidence:** 3

**Summary:**

In this work, the authors consider the problem of Federated linear bandits with finite adversarial actions. This is the first time investigating a setting where federated clients are faced with a set of actions to choose from that changes over time in an oblivious adversarial manner, and the authors of this paper do so for  both synchronous and asynchronous versions of the problem.
To handle this problem, the authors propose two algorithms that are based on SupLinUCB (Chen et al. 2011), which is a standard approach for linear bandits and adapted to federated bandits. Then, they combine this S-LUCB approach with two different protocols to tackle both the asynchronous and the synchronous version of the problem.One of the challenges of federated learning comes with the cost of the communication between the clients and the server.
In the asynchronous setting, only one client is active at the time and the the synchronization rate depends on each individual client. For this problem, the authors prove that the proposed algorithm achieves an order optimal (up to log factors) high probability bound of order $O(\sqrt{dT})$ while ensuring that their total communication cost is logarithmic in T.
In the synchronous case, layers of clients work simultaneously which allows for the server to take advantage of the various information provided by the clients. This means that these clients don't just synchronize when the information that they have has changed, but they also do so periodically to ensure that their information is up to date. This extra information provided by the  server to the clients allows to actually lower the total number of communications and obtain a $\tilde O(\sqrt{dMT_c})$ regret bound while keeping a time independent bound on the number of communication.

The authors provide detailed proofs of their results in the appendices and experiments, both on generated and real life datasets to highlight how the regret and the communication costs evolve in terms of arrival patterns and number of clients.




**Strengths:**

This works studies a new variant of the federated linear bandits problem where the clients are facing finite adversarial arm sets instead of either finite fixed armed sets or infinite arm sets.
To do so, they build upon existing algorithms for linear bandits with an adversarial arm set, SupLinUCB (Chu et al. 2011) and for asyncronous federated learning FedLinUCB (He et al. 2022a). The novelty of their approach mainly resides in the improved communication protocol, in particular in the synchronous setting, where they take advantage of a layer structure to limit the communication cost while still synchronizing regularly.
In both the asynchronous and the synchronous settings, they recover the regret of  SupLinUCB in the single player setting (Chu et al. 2011), and are within a log T factor of the state of the art results for federated synchronous and asynchronous learning, both in terms of regret and communication costs.

Overall, this is a well presented paper that provides some interesting results for a new variation of the federated learning problem, which might be more relevant in practice, which is reinforced by the fact that they also run experiments on real worlds datasets.
Presenting results for the corrupted setting is also a nice generalization of their results.


**Weaknesses:**

In the experiments section, it would be interesting to see the proposed algorithms compared with other results for federated learning: The results in Appendix H are promising and could benefit from being extended.
It  would also be nice if the legends on the plot could be larger so it is a bit easier to read.


**Questions:**

The problems studied in federated learning are very close to these of multiplayer multiarmed bandits. Have you thought of whether there are results that could bridge the two settings?

---

> ### Author Rebuttal · Authors · 2023-08-08
>
> We appreciate that the reviewer liked our federated linear bandit model, as well as for providing a thoughtful summary of our paper.
>
> **Q1**: "In the experiments section, it would be interesting to see the proposed algorithms compared with other results for federated learning..."
>
> **Response**: We thank the reviewer for the suggestion. In the main paper, we proposed algorithms that provably achieve nearly minimax optimal regret while maintaining a small amount of communication cost. In the experiments section, we aim to have a better interpretation of the proposed algorithms by empirically studying the impact of the arrival pattern, the number of clients, and the trade-off between regrets and communication costs. In other words, gaining intuition and bringing the interpretation of the proposed algorithm is our goal in the setup of the experiments.
> To this end, we did not conduct the experiments for other existing algorithms, which as far as we know are not minimax optimal in our setting.
>
> **Q2**: "The problems studied in federated learning are very close to these of multiplayer multiarmed bandits. Have you thought of whether there are results that could bridge the two settings?"
>
> **Response**: A majority of the multiplayer multi-armed bandits settings focus on handling the collision caused by multiple players pulling the same arm, while in the federated bandits settings, the multiple players communicate with a central server to better solve their own local decision-making problem by the information acquired from other players. We thank the reviewer for the insightful observation regarding the existing research in the domain of multi-player and multi-armed bandit problems, we are not aware of any work combining the study of the communication costs and the collisions.

---

> > ### Comment · Area_Chair_AACD · 2023-08-17
> > **Please engage in the rebuttal**
> >
> > Dear reviewer,
> > Please acknowledge the author's response and tell us if the replies changed your assessment.

---

### Official Review · Reviewer_5B2X · 2023-07-08

**Soundness:** 2 fair
**Presentation:** 2 fair
**Contribution:** 2 fair
**Rating:** 4
**Confidence:** 4

**Summary:**

This paper addresses the linear bandits problem in the context of federated learning. It proposes a general algorithm called FedSupLinUCB that solves a linear contextual bandit problem with finite adversarial action sets that may vary across clients. The paper also considers practical challenges in federated settings such as synchrony, communication efficiency, and time-evolving scenarios.

**Strengths:**

This paper explores the linear contextual bandit problem with finite adversarial action sets in the federated learning setting. Additionally, it addresses several practical challenges in federated settings, such as synchrony, communication efficiency, and time-evolving scenarios.

**Weaknesses:**

Background and challenge:

- The introduction and preliminaries do not provide a detailed definition of the time-evolving and adversarial arm. To improve comprehension, it would be helpful to explain the impact of these factors on the federated linear bandit problem.
- Although the paper discusses the main challenges of federated linear bandits, it should emphasize the key differences from traditional linear bandit problems, such as single-player bandits or distributed bandits.
- The paper lacks strong motivation due to the vague background and challenge.

Novelty:

- The main theoretical contributions of the paper extend previous work on linear bandit problems to the federated setting. For example, FedSumLinUCB is an extension of SupLinUCB and OFUL. Robust Async-FedSupLinUCB incorporates ideas from He et al. (2020b).
- The theory and methodology employed in the paper appear to be borrowed from the literature.

Experiment setting:

- The reward function utilized in the experiments, where $r_{t, a}=\theta^{\top} x_{t, a}+\epsilon_t$ , does not align with the settings described in the paper. To my knowledge, it is more general to let the reward function different across clients with certain variances according to the environments.
- The experiments conducted in the paper are limited. Only a classic algorithm proposed in 2011 by Chu et al. is used as a baseline, and not all proposed algorithms are fully evaluated.

Writing:

- It is unusual for a conference paper to have 10 sections.

**Questions:**

- What are the specific challenges in the federated linear bandit problem? Federated learning was originally proposed to address data-sharing issues. Does the federated linear bandit problem face similar challenges related to system efficiency and heterogeneity?
- What unique solutions or methods does the paper contribute? It appears that some challenges have already been addressed by previous works.

**Limitations:**

No limitations are discussed in the main paper.

---

> ### Author Rebuttal · Authors · 2023-08-08
>
> We thank the reviewer for the interesting questions regarding the proposed FedSupLinUCB algorithm.
>
> **Q1**: "The introduction and preliminaries do not provide a detailed definition of the time-evolving and adversarial arm. To improve comprehension, it would be helpful to explain the impact of these factors on the federated linear bandit problem."
>
> **Response**: We appreciate the reviewer's attention to the setting of the contexts set. In Section 3.1 (line 110), we have provided a clear definition of the time-evolving context sets, indicating that the context sets change over time and are different among clients. The inclusion of both finite and adversarial context sets in our problem formulation presents a more challenging scenario compared to the Fed-PE algorithm, which specifically handles finite and fixed contexts. For that situation, techniques like G-optimal design are applicable, though do not apply in our formulation. To address the challenges posed by time-evolving contexts, we have introduced the FedSupLinUCB framework. We will highlight the distinctiveness when presenting the results and comparisons.
>
> **Q2**: "Although the paper discusses the main challenges of federated linear bandits, it should emphasize the key differences from traditional linear bandit problems, such as single-player bandits or distributed bandits."
>
> **Response**: In our study, we concentrate on a federated linear bandit model. In comparison to single-player bandit and distributed bandit approaches, we leverage the benefit of data sharing through communication between the clients and the central server. Our main focus is on achieving optimal regret while keeping communication costs low. Specifically, in our model, we consider a star-shaped structure, where M clients interact with a central server to expedite the decision-making process. Additionally, we highlight the benefits of the federated linear bandit framework in the remarks following the theorem in our paper.
>
> It's worth noting that there does not appear to be a clean separation between "distributed bandits" and "federated bandits". Nevertheless, we contend that the "federated bandits" setting enables a more comprehensive modeling of the communication and computation, and can be viewed as focusing more on the impact of different communication and computation modes, which is indeed the perspective we take in this work.
>
> **Q3**: "The paper lacks strong motivation due to the vague background and challenge."
>
> **Response**: Linear bandits with adversarial and finite actions find numerous applications, including recommendation systems [R1]. The distributed nature of these applications naturally aligns with the setting we have studied in this paper, as we have also mentioned in the introduction section of our paper.
>
> **Q4**: "The main theoretical contributions of the paper extend previous work on linear bandit problems to the federated setting...The theory and methodology employed in the paper appear to be borrowed from the literature."
>
> **Response**: Similar to most of the previous works on federated bandits, whose technical aspects resemble the counterparts of single-player bandits, we also utilize techniques in single-player bandits as analysis tools for this specific federated linear bandit problem. However, we believe that the study of the federated bandit problem is important and valuable, particularly due to the recent trend of invoking more edge computing resources. To address the specific challenges in the federated setting, we proposed the FedSupLinUCB framework, where the Async-FedSupLinUCB and Sync-FedSupLinUCB both achieve near-optimal regrets while significantly reducing communication costs.
>
> **Q5**: "The reward function utilized in the experiments, where $r_{t,a_t} = \theta^{\top} x_{t,a_t}^i + \epsilon_t$, does not align with the settings described in the paper. The experiments conducted in the paper are limited..."
>
> **Response**: We conducted our experiment to elaborate on the theoretical analysis and indeed aligned it with the problem formulation presented in Section 3.1. We made the assumption that all clients share the same underlying parameter $\theta$. This assumption allows us to leverage the sharing of information among clients, thereby benefiting the decision-making process of each individual client, and this is the fundamental reason that federated learning is able to gain performance improvement over learning by individual agents. Considering our assumptions regarding finite and adversarial context sets, we conduct a comparison with the order-optimal benchmark, SupLinUCB, to showcase the advantages of our federated framework.
>
> **Q6**: "What are the specific challenges in the federated linear bandit problem? Federated learning was originally proposed to address data-sharing issues. Does the federated linear bandit problem face similar challenges related to system efficiency and heterogeneity? What unique solutions or methods does the paper contribute? It appears that some challenges have already been addressed by previous works."
>
> **Response**: The major technical challenges (also for almost all the federated bandits works) are to incorporate the techniques in the single-player algorithms into the federated scenario while explicitly considering the impact of communication cost. Yet the focus on the communication cost under different update and computation modes led us to the proposed new algorithms in this work. We believe the study of such federated contextual bandits problem is important and valuable, and the proposed algorithms and their analysis advance our understanding and provide insights into the problem in terms of maintaining the near-optimal regret while significantly mitigating the communication costs.
>
> [R1] Ruan, Yufei, Jiaqi Yang, and Yuan Zhou. ”Linear bandits with limited adaptivity and learning distributional optimal design.” Proceedings of the 53rd Annual ACM SIGACT Symposium on Theory of Computing. 2021.

---

> > ### Comment · Area_Chair_AACD · 2023-08-17
> > **Please engage in the rebuttal**
> >
> > Dear reviewer,
> > Please at least acknowledge the author's response and ideally explain if and why their reply does not change your mind.

---

> > ### Comment · Reviewer_5B2X · 2023-08-19
> > **Thanks for the feedback**
> >
> > Thank you for addressing my concerns. However, I remain uncertain, especially in terms of the core challenges in FL like client data heterogeneity, the distinct difference between distributed bandit and federated bandit, and the paper's singular contributions beyond the amalgamation of techniques from two distinct domains.

---

### Comment · Area_Chair_AACD · 2023-08-18
**Discussion overview**

Dear reviewers and authors,

From the rebuttal, I take away that there are no strong disagreements between authors and reviewers on any factual matters in the paper. The main reason the scores are not higher, is that the reviewers are not convinced of the significance and novelty of the proposed algorithmic solution, correct?

---

### Decision · Program_Chairs · 2023-09-21

**Decision:**

Accept (poster)

**Comment:**

This paper studies the problem of Federated linear bandits with finite adversarial actions. They adapt the classical SupLinUCB algorithm to work in the federated learning setup.

There is a general consensus among reviewers that this is a correct and solid work, however, the overall novelty is limited (contribution score between 2 and 3). The main strength mentioned is that the regret bounds are close to optimal, but neither the algorithm nor the analysis are seen as being especially challenging.
Additionally, this work provides several minor contributions such as the variance-adaptive and the corruption setting.
Overall, this paper is a complete package with several contributions, though none of its own of high impact. Three of four reviewers see this above the acceptance threshold.
The final reviewer is expressing a concern that the studied setting misses subtleties of the original federated learning framework and is more closely related to distributed bandits. However, there is a long record of FL work that uses the assumptions of this paper.